TOPICAL REVIEW

# Neuromotor changes in postural control following bed rest

Ramona Ritzmann[1,2], Christoph Centner[1] , Luke Hughes[3] , Janice Waldvogel[1]
and Uros Marusic[4,5]

[1] *Department of Sport and Sport Science, University of Freiburg, Freiburg, Germany*
[2] *Innovation Translation Center, AO Foundation, Davos, Switzerland*
[3] *Faculty of Health & Life Sciences, Northumbria University, Newcastle, UK*
[4] *Institute for Kinesiology Research, Science and Research Centre Koper, Koper, Slovenia*
[5] *Department of Health Sciences, Alma Mater Europaea University, Maribor, Slovenia*

Handling Editors: Richard Carson & Matthew Fogarty

The peer review history is available in the Supporting information section of this article (https://doi.org/10.1113/JP285668#support-information-section).

**Abstract figure legend** Prolonged bed rest (BR) (both horizontal and head-down tilt) leads to neuromotor adaptations that impair postural control. Structural and functional changes occur in the brain, including a reduction in grey and white matter volume, alterations in functional connectivity, and shifts in cerebrospinal fluid distribution. At the spinal level, prolonged BR reduces spinal excitability, delays the latency of the stretch reflex and alters motor control mechanisms. Sensory integration is also disturbed, which manifests itself in impaired proprioceptive feedback, vestibular dysfunction and reduced visual acuity and contrast sensitivity.

**Ramona Ritzmann,** PhD, specializes in neurophysiological stimulation techniques to study motor control and neuroplasticity. She has worked with the European Space Agency on gravity and deconditioning research and led biomechanics and performance diagnostics at a Swiss sports medicine clinic. Since 2021, she has been with the AO Foundation, becoming Head of Clinical Operation in 2022, overseeing clinical research and trials. **Uros Marusic** holds a B.S. in Biomedical Engineering and a Ph.D. in Kinesiology. He is the Head of the Laboratory for Kinesiological Research and leads the Slovenian Mobile Brain/Body Imaging Lab (SloMoBIL) at ZRS Koper. His research focuses on muscle-brain crosstalk, particularly in aging and neurodegeneration (e.g., Parkinson's disease), with implications for neurophysiology in space.

The Journal of Physiology

**Abstract** Chronic bed rest (BR) serves as a model for studying the effects of prolonged immobility on physiological and neuromotor functions, particularly postural control. Prolonged BR leads to significant deconditioning of postural balance control, characterized by increased sway path lengths, sway velocity and fall risk, independent of muscle strength. These changes are linked to neural adaptations at spinal and supraspinal levels, including structural and functional brain changes, such as alterations in grey and white matter, increased cerebellar activation, reduced spinal excitability and increased latencies within reflex circuitries. Additionally, BR disrupts sensory integration from proprioceptive, visual and vestibular systems, impairing postural stability. Visual reliance remains stable during BR, though decreased visual acuity and contrast sensitivity are noted. Moreover, BR-induced shifts in cerebrospinal fluid contribute to altered brain activity, impacting sensorimotor function. Vestibular system adaptations, including changes in vestibulospinal reflexes, further exacerbate balance impairments. Understanding these mechanisms is crucial for developing interventions to mitigate the adverse effects of BR on postural control and prevent prolonged recovery times or increased risk of injury. This review highlights the need for further research into the neural underpinnings of BR-induced postural instability, with a focus on sensory integration and neuroplasticity.

(Received 31 August 2024; accepted after revision 5 March 2025; first published online 15 April 2025)

**Corresponding author** R. Ritzmann: University of Freiburg, Sandfangweg 4, 79111 Freiburg, Germany. Email: ramona.ritzmann@uni-freiburg.de

## Introduction

Chronic bed rest (BR) is often used as a model for studying the effects of prolonged immobility and has been shown to result in significant physiological and neuromechanical adaptations that affect various bodily functions (Pišot et al., 2016; Schefold et al., 2020). Among these adaptations, deteriorations in postural balance control and postural equilibrium are particularly critical because they are essential for human mobility and stability (Saumur et al., 2020). The physical deconditioning and postural damage resulting from prolonged BR persists beyond the acute phase of inactivity and is associated with prolonged recovery times (Pavy-Le Traon et al., 2007). This phenomenon has gained increasing socioeconomic importance and has triggered a scientific debate on the timing, consequences and mechanisms underlying these sensorimotor deficits (Marusic, Narici et al., 2021). Importantly, the effects of prolonged BR can be even more pronounced in older people (Pišot et al., 2016), who are already at increased risk of falls, muscle atrophy and prolonged hospitalization as a result of pre-existing comorbidities (Kehler et al., 2019).

Physical disuse significantly impairs postural balance, with the extent of impairment dependant on the duration of BR and the complexity of the postural task (Saumur et al., 2020). The longer the BR duration and the more demanding the postural task, the more pronounced the degradative adaptation processes (Miller et al., 2018; Sarabon & Rosker, 2015; Stuempfle & Drury, 2007). Interestingly, functional and morphological muscle alterations such as muscle atrophy or declines in strength do not appear to be associated with increases in static postural instability following BR experiments (Sarabon & Rosker, 2013). Thus, resistance training countermeasures that effectively maintain lower limb volume and strength do not appear to affect postural stability (Haines, 1974; Kouzaki et al., 2007). This missing association highlights that impaired postural balance control after BR cannot be attributed to loss of strength alone, but is attributed to altered sensorimotor function, including changes in spinal and supraspinal circuits (Mackenzie et al., 2024; Ritzmann et al., 2018). In older people in particular, these deficits can worsen as a result of the age-related decline in proprioceptive and vestibular functions, which further increases susceptibility to falls after BR. Numerous studies have assessed the magnitude of peripheral and central changes following BR (Koppelmans et al., 2017; Liao et al., 2012; Muir et al., 2011; Yuan, Koppelmans, Reuter-Lorenz, de Dios, Gadd, Riascos et al., 2018) but there are several important concepts of BR experiments that need to be considered to interpret the effects on postural balance control.

In the previous literature, for example, BR studies are generally used as a surrogate paradigm to mechanical unloading during spaceflights. Because weightlessness has been shown to induce considerable fluid shifts to the upper body (Thornton et al., 1987), BR studies are often coupled with a head-down tilt position to mimic conditions during microgravitational manoeuvres. Interestingly, head-down tilt positions have been demonstrated to influence neural markers such as electrocortical activity (Brauns et al., 2021; Koppelmans et al., 2017) and thus need to be considered in a holistic interpretation of

potential mechanisms associated with declines in postural balance control.

Understanding the mechanisms behind these neuromotor changes is crucial for the development of targeted interventions aiming to mitigate the adverse effects of BR on postural balance control. The aim of this review is to determine the effects of BR on balance control and elucidate the underlying mechanisms responsible for these changes (Fig. 1). Starting from the time-dependent (short-term: <2 weeks, medium-term: 3–5 weeks and long-term: ≥6 weeks) functional adaptations presented in the first subsection using biomechanical parameters, we then consider the adaptations of neuronal correlates at the spinal and supraspinal level. First, structural, functional and brain electrocortical adaptations are documented and related to their implications for sensorimotor function and balance control. Second, adaptations in neuronal plasticity and spinal motor control are described and related to biomechanical parameters of balance control. Third, changes in proprioception, vestibular and visual cues as a result of chronic BR are summarized and their significance in relation to balance control is discussed.

## Biomechanics of declined postural balance control

To scientifically differentiate functional adaptations and understand the neuromotor cause-and-effect mechanism, adaptation of postural balance control to BR has been investigated in measurement paradigms of static (Table 1)

(de Martino et al., 2021, 2022; Koppelmans et al., 2017; Kouzaki et al., 2007; Mulavara et al., 2018; Muir et al., 2011; Mulder et al., 2014; Ritzmann et al., 2018; Viguier et al. 2009), dynamic (Table 2) (Clement et al., 2015; Haines, 1974; Koppelmans et al., 2017; Tays et al., 2022; Viguier et al., 2009) and anticipatory-reactive conditions (Table 3) mimicking events of falls (Miller et al., 2018; Sarabon & Rosker, 2015; Šarabon et al., 2018) concomitantly with variation of experimentally manipulated sensory information. Sensory manipulation includes visual feedback with comparisons of eyes open and eyes closed conditions (de Martino et al., 2022; Koppelmans et al., 2015, 2017; Mulder et al., 2015; Ritzmann et al., 2018; Viguier et al., 2009), combined with altered proprioceptive input by unstable surfaces (e.g. foam pads) (Clement et al., 2015; Mulder et al., 2014) or by using a sway-referenced support surface (Koppelmans et al., 2015, 2017; McGregor et al., 2023; Miller et al., 2018; Mulavara et al., 2018; Paloski et al., 2017; Reschke et al., 2009; Tays et al., 2022; Yuan et al., 2018) and/or a rigged vestibular system using dynamic head tilting (Mackenzie et al., 2024; Miller et al., 2018; Mulder et al., 2015).

The biomechanical consequences of postural balance control undergo significant deconditioning with BR, characterized by up to 50% prolonged sway path lengths of the centre of pressure and centre of force with ≥60 days of BR (Saumur et al., 2020), a doubled and tripled sway velocity after 60 days and 90 days of BR during static conditions (Kouzaki et al., 2007;

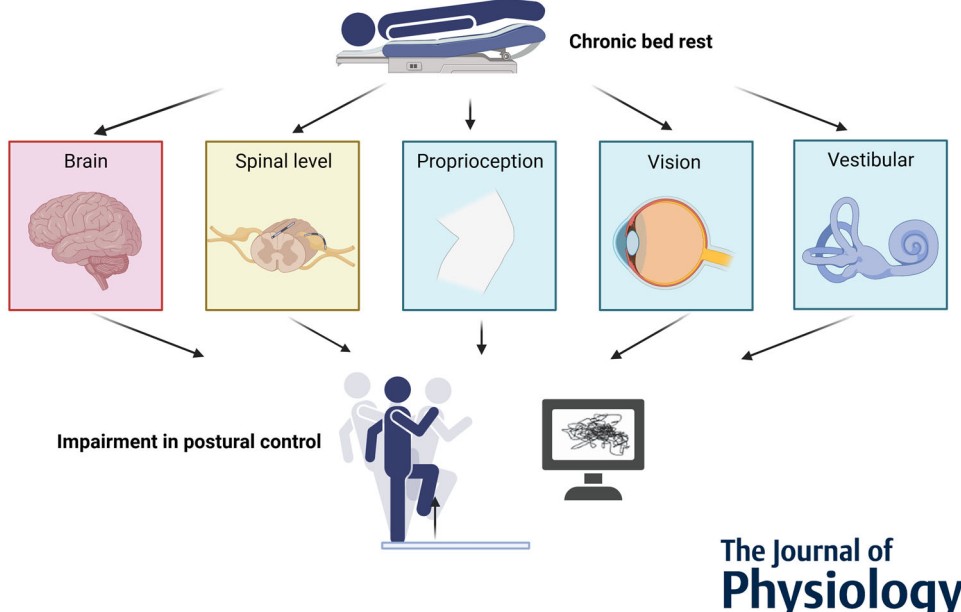

**Figure 1. Neuromotor adaptations to bed rest: focus on postural control**
The concept of the manuscript is based on the neuromotor adaptations to chronic BR with reference to postural control. The focus is on the adaptations of brain structure and function, reflex control at the spinal level, and the impairment of sensory sources for movement and stance control.

**Table 1. Static balance condition**

| Reference (year) | Bed rest duration (days) | Type of bed rest | Sample size (n) (male/female) | Age (years) | Methods deployed to assess balance control | Measured variable | Adaptation of balance control after compared to before bed rest |
|---|---|---|---|---|---|---|---|
| Koppelmans et al. (2017) | 70 | Head down tilt bed rest | 18 (18/0) | 31.1 ± 4.7 | Sensory Organization Test: Static balance in quiet both leg stance, 20 s eyes closed, with and without dynamic head tilt | Continuous equilibrium score computed by COP deviation before and 0, 7 and 12 days after re-ambulation | Reduced continuous equilibrium score at day 0 after bed rest for conditions with and without head tilt. No effects 7 and 12 days after bed rest. |
| Muir et al. (2011) | 60 and 90 | Head down tilt bed rest | 13 (8/5) | 35.6 ± 7.1 | Static balance in quiet leg stance, 4 min, eyes open and eyes closed | COP sway path and peak velocity in medio-lateral and anterior-posterior direction, and power spectral density analysis, stabilogram diffusion analysis before and after bed rest | COP displacement and peak velocity increased in medio-lateral and anterior-posterior direction for eyes open and eyes closed conditions. Increased mid- and high-frequency spectral density and stabilogram diffusion 60 and 90 days after compared to before bed rest for eyes open and closed conditions. |
| Viguier et al. (2009) | 60 | Head down tilt bed rest | 8 (0/8) | 34 ± 1 | Static balance in quiet both leg stance, 51 s, eyes open and eyes closed | COP sway path before and 1, 4 and 10 days after re-ambulation | Total COP displacement increased in eyes open and closed condition at 1 and 4 days after compared to before bed rest. No effects 10 days after bed rest. |

*(Continued)*

**Table 1. (Continued)**

| Reference (year) | Bed rest duration (days) | Type of bed rest | Sample size (*n*) (male/female) | Age (years) | Methods deployed to assess balance control | Measured variable | Adaptation of balance control after compared to before bed rest |
|---|---|---|---|---|---|---|---|
| Ritzmann et al. (2018) | 60 | Head down tilt bed rest | 11 (11/0) | 28 ± 6 | Static, quiet monopedal stance with eyes open and eyes closed. 2 × 10 s | COF displacement, COF sway velocity, dominant frequency and the standard ellipse area (90% movement area) in medio-lateral and anterior-posterior direction were calculated for before and 0, 7, 13, 28 and 90 days after bed rest in eyes open and closed conditions | COF displacement, and standard ellipse area in medio-lateral and anterior-posterior direction for eyes open and eyes closed conditions increased 1 and 7 days after bed rest. COF velocity increase up to 13 days after bed rest for eyes open and closed conditions. No changes were found in dominant frequency. |
| Mulavara et al. (2018) | 70 | Head down tilt bed rest | 10 (10/0) | 38 ± 7 | Sensory Organization Test: Static balance in quiet both leg stance, 3 × 20 s, eyes closed with the head erect and eyes closed with dynamic head tilts | Continuous equilibrium score computed by COP deviation before and after re-ambulation | Reduced continuous equilibrium score after compared to before bed rest. |

*(Continued)*

**Table 1. (Continued)**

| Reference (year) | Bed rest duration (days) | Type of bed rest | Sample size (n) (male/female) | Age (years) | Methods deployed to assess balance control | Measured variable | Adaptation of balance control after compared to before bed rest |
|---|---|---|---|---|---|---|---|
| Mulder et al. (2014) | 5 | Head down tilt bed rest | 10 (10/0) | 29.4 ± 5.9 | Static balance in quiet both leg stance on a foam pad, 3 × 30 s, eyes open with the head erect, eyes closed with the head erect and eyes closed with dynamic head tilts | Equilibrium score calculated with COM sway angles derived from anterior-posterior and medial-lateral COP data before, 0 and 3 days after bed rest | No differences in the Equilibrium scores were observed between before and after bed rest neither with the eyes open nor with the eyes closed. Equilibrium scores decreased for 0 and 3 days after bed rest when eyes closed while dynamic head tilting. |
| De Martino et al. (2021) | 60 | Head down bed rest | 8 (6/2) | 34 ± 8 | Static balance in quiet both leg stance, 70 s, eyes open, eyes closed, eyes open on foam and eyes closed on foam | Total COP sway path, peak sway velocity and sway frequency power in medio-lateral and anterior-posterior direction before and after bed rest | In all four conditions, COP displacement and peak sway velocity increased in medio-lateral and anterior-posterior direction Increased frequency power in anterior-posterior plane |
| Kouzaki et al. (2007) | 20 | Head down bed rest | 6 (6/0) | 23.3 ± 4.9 | Static balance in quiet both leg stance, 5 × 40 s, eyes open and eyes closed | COP mean sway velocity was calculated before and immediately after bed rest | COP mean velocity increased in both, eyes open and eyes closed, condition. |

*(Continued)*

**Table 1. (Continued)**

| Reference (year) | Bed rest duration (days) | Type of bed rest | Sample size (n) (male/female) | Age (years) | Methods deployed to assess balance control | Measured variable | Adaptation of balance control after compared to before bed rest |
|---|---|---|---|---|---|---|---|
| Sarabon & Rosker (2013) | 14 | Horizontal bed rest | 16 (16/0) | 59.6 ± 3.4 | Static balance in quiet both leg stance, 3 × 60 s, eyes open | COP displacement and frequency in medio-lateral and anterior-posterior direction rambling-and trembling sway decomposition before, 0 and 14 days after bed rest | COP displacement for rambling and trembling decomposition increased in medio-lateral and anterior-posterior direction immediately after compared to before bed rest. Rambling increased more in frequency (anterior-posterior), while trembling increased more in amplitude. After 14 days after bed rest parameters returned to the values before bed rest. |
| Šarabon et al. (2018) | 21 | Horizontal bed rest | 14 (14/0) | 26 ± 5 | Static balance in quiet both leg stance, with eyes open, parallel stance with eyes closed, and semi-tandem stance with eyes open and eyes closed | Average COP velocities, amplitudes and frequencies in medio-lateral and anterior-posterior direction before and immediately after bed rest | Increase in average COP velocities, amplitudes and frequencies in medio-lateral and anterior-posterior direction immediately after bed rest compared to before bed rest. |

*(Continued)*

**Table 1. (Continued)**

| Reference (year) | Bed rest duration (days) | Type of bed rest | Sample size (*n*) (male/female) | Age (years) | Methods deployed to assess balance control | Measured variable | Adaptation of balance control after compared to before bed rest |
|---|---|---|---|---|---|---|---|
| Clément et al. (2015) | 5 | Head down tilt bed rest | 10 (10/0) | 34.2 ± 2.1 | Static balance in quiet both leg stance on foam pad, 3 × 30 s, eyes open with head erect, eyes closed with head erect and eyes closed with dynamic head tilts | Equilibrium score was calculated with COM sway angles derived from anterior-posterior and medial-lateral COP data before, 0 and 3 days after bed rest | No significant differences in the Equilibrium scores were observed between before and after bed rest neither with the eyes open nor with the eyes closed with head erect or eyes closed while dynamic head tilting |
| Tays et al., 2022 | 60 | Head down tilt bed rest | 24 (16/8) | 33.3 ± 9.2 | Sensory Organization Test: Static balance in quiet both leg stance, 3 × 20 s, eyes closed on sway referenced platform and eyes closed with dynamic head tilts | Equilibrium score was calculated from peak-to peak excursion of the COM before, 0 and 10 days after bed rest | Equilibrium scores decreased 0 days after compared to before bed rest. 10 days after bed rest equilibrium score significantly increased indicating a recovery from balance deficits. |

*(Continued)*

**Table 1. (Continued)**

| Reference (year) | Bed rest duration (days) | Type of bed rest | Sample size (n) (male/female) | Age (years) | Methods deployed to assess balance control | Measured variable | Adaptation of balance control after compared to before bed rest |
|---|---|---|---|---|---|---|---|
| Mackenzie et al. (2024) | 60 | Head down tilt bed rest | 18 (18/0) | 34 ± 9 | Static balance in quiet both leg stance, 40 s, with three head orientations and eyes open and eyes closed | COP mean sway velocity and COP area, which was defined as a 95% confidence ellipse fitted to the COP data, were calculated | After Bed rest a significant increase in postural sway was shown for both, eyes open and eyes closed condition, however increase was higher when eyes were closed. Significant increase was detected for after bed rest compared to before bed rest in eyes open and eyes closed condition. 6 days after bed rest the values returned to the baselines before bed rest. |
| Miller et al. (2018) | 70 | Head down tilt bed rest | 10 (10/0) | 37.7 ± 7.2 | Static balance in quiet both leg stance, 6 × 20 s, pitch heads ± 20° | Continuous equilibrium score calculated 0, 1, 6, 12 and 30 days after bed rest | Reduced continuous equilibrium score 0 days after compared to before bed rest. No effects were shown for days 1 to 30 after bed rest. |

Abbreviations: COF, centre of force; COM, centre of mass; COP, centre of pressure COP.

**Table 2. Dynamic balance condition**

| Reference (year) | Bed rest duration (days) | Type of bed rest | Sample size (n) (male/female) | Age (years) | Methods deployed to assess balance control | Measured variable | Adaptation of balance control after compared to before bed rest |
|---|---|---|---|---|---|---|---|
| Koppelmans et al. (2017) | 70 | Head down tilt bed rest | 18 (18/0) | $31.1 \pm 4.7$ | Functional mobility test (FMT) including obstacles | Times to finish the FMT at 0, 7 and 12 days after re-ambulation | Increased time to finish the FMT 0 and 7 days after re-ambulation. No effects 12 days after bed rest |
| Tays et al., 2022 | 60 | Head down tilt bed rest | 24 (16/8) | $33.3 \pm 9.2$ | FMT including obstacles | Times to finish the FMT at 7 days before, 0 and 10 days after bed rest | Increased time to finish the FMT 7 days before and immediately after bed rest. No effects 10 days after bed rest |
| Clément et al. (2015) | 5 | Head down tilt bed rest | 10 (10/0) | $34.2 \pm 2.1$ | dynamic gait test consisting of eight conditions: (1) walking 6 m on a level surface at normal pace; (2) while changing gait speed at command; (3) while looking to the right or to the left upon command; (4) or while looking up or down; (5) while turning as quickly as possible to face the opposite direction and stop; (6) while stepping over an obstacle; (7) while turning around cones; and (8) walking up and down stairs | Dynamic gait index (DGI) was calculated to measure the performance of each dynamic gait test condition. Index was rated by an experienced rater (rating from 0 to 3 scores according to criteria) 1 day before and 3 days after bed rest | Immediately after bed rest DGI was lower by one point in half of the subjects, especially when subjects had to look sideways or up and down or when walking up and down stairs. DGI scores were not significantly different. |

*(Continued)*

**Table 2. (Continued)**

| Reference (year) | Type of bed rest | Bed rest duration (days) | Sample size (n) (male/female) | Age (years) | Methods deployed to assess balance control | Measured variable | Adaptation of balance control after compared to before bed rest |
|---|---|---|---|---|---|---|---|
| Haines (1974) | Horizontal bed rest | 14 | 7 | 20.4 ± 1.3 | Balance performance scores executed in conditions with open and closed eyes including floor line walk, left-and right leg rail balance test, sharpened Romberg with and without rail and rail walk | Floor line walk and rail walk was scored in distance and time, left-and right leg rail balance test was scored in time to fall, sharpened Romberg with and without rail was scored in time to fall | Increased time and reduced distance for floor line walking with eyes closed and rail walking with eyes open after compared to before bed rest with return to baseline after 5 days after bed rest Left leg rail balance test revealed a reduced time to failure up to 5 days after bed rest compared to before bed rest |
| Viguier et al. (2009) | Head down tilt bed rest | 60 | 8 | 34 ± 1 | Upright quiet both leg stance for 51 s on a platform moveable either in anterior-posterior or medio-lateral direction. Eyes open and eyes closed condition | COP sway path before and 1, 4 and 10 days after re-ambulation | COP sway path increased after bed rest in anterior-posterior and medio-lateral direction. Degradation of balance is most pronounced after bed rest and progressively recovers until 10 days after bed rest. Antero-posterior-performance degrades more than medio-lateral. Eyes closed conditions show stronger degradation in balance control compared to eyes open condition. |

Abbreviations: DGI, dynamic gait index; FMT, functional mobility test.

**Table 3. Anticipatory postural reactions mimicking falls**

| Authors (year) | Bed rest duration (days) | Type of bed rest | Sample size (*n*) (male/female) | Age (years) | Methods deployed to assess balance control | Measured variable | Adaptation of balance control after compared to before bed rest |
|---|---|---|---|---|---|---|---|
| Sarabon and Rosker (2015) | 14 | Horizontal bed rest | 16 (16/0) | 59.6 ± 3.4 | Evaluation of anticipatory postural adjustments and postural reflex responses of the abdominal wall and back muscles | 0 and 14 days after bed rest | Immediately after the bed rest latencies of anticipatory postural adjustments showed significant shortening, near to complete recovery was reached 14 days after bed rest. Reactive response latencies increased from pre-bed-rest to 0 and 14 days post-bed-rest outlasting the 14-day post-bed-rest rehabilitation. |
| Miller et al. (2018) | 70 | Head down tilt bed rest | 10 (10/0) | 37.7 ± 7.2 | Recovery from fall to stand test instructing the subjects to lie down and regain stable upright stance as quickly as possible | Time to stand stable and COP mean sway speed 0, 1, 6, 12 and 30 days after bed rest | Time to stand stable was increased after compared to before bed rest. COP mean sway speed was increased up to 6 days after bed rest and reached baseline afterwards. |
| Šarabon et al. (2018) | 21 | Horizontal bed rest | 14 (14/0) | 26 ± 5 | Trunk stabilization after perturbation including anticipatory postural adjustments and postural reflex responses | Latencies and amplitudes of postural adjustments before and immediately after bed rest | No changes in anticipatory postural adjustments or postural reflex response latencies in any of the conditions. Trunk muscle amplitude and rise in muscle activity decreased after bed rest. |

Muir et al., 2011), a 10–40% increased sway frequency during static postural balance tasks (Sarabon & Rosker, 2013) and changes in kinetic parameters (Sarabon & Rosker, 2015) different from normative values of healthy and mobile control measures before BR. Studies have demonstrated increased sway path lengths in both the medio-lateral and antero-posterior directions, indicative of diminished static postural balance control (Dupui et al., 1992; Ritzmann et al., 2018). There is a notable shift from ankle-dominated strategies to increased reliance on hip movements to maintain balance (Clément et al., 2015; de Martino et al., 2022; Ritzmann et al., 2018), which is probably a proximal compensatory mechanism as a result of modulated sensory feedback in the lower extremities, the eyes and the vestibular organ (Clément et al., 2015; Mackenzie et al., 2024; Yuan, Koppelmans, Reuter-Lorenz, de Dios, Gadd, Riascos et al., 2018). Kinematic analyses show exaggerated joint angles and limb movements, reflecting adaptations to maintain stability under deconditioned states (Sarabon & Rosker, 2015) and, most worryingly, a higher fall prevalence (in experimental conditions) (Miller et al., 2018) resulting in an increased risk of moderate and severe injuries (Stuempfle & Drury, 2007).

## Adaptation of neuronal correlates to bed rest

Several neuronal factors account for the effects of disuse on postural equilibrium. These include neuroplasticity on the spinal and supraspinal level and maladaptive interpretation of sensory feedback delivered by vestibular, visual and mechanosensitive sources (Fig. 1). Compared with functional assets, neural aspects linked to static and dynamic postural deconditioning effects of disuse have received less attention in the past decades. To address these omissions, recently published studies have used imaging procedures (Koppelmans et al., 2017), myograms (Paloski et al., 2017) and neurophysiological stimulation methods applied to the brain (Miyazaki et al., 2002) and peripheral nerves (Paloski et al., 2017; Yamanaka et al., 1999) in combination with traditional functional diagnostics to establish causal links between neuronal correlates and worsened balance control after periods of chronic disuse (Fig. 2).

**Structural, functional and brain electrocortical adaptations.** Prolonged BR and head-down tilt BR (HDTBR) have been shown to cause significant changes in the brain that negatively affect sensorimotor function and postural balance control. Direct studies on the structural and functional characteristics of the brain in relation to postural balance control are limited, but the following three subsections discuss these adaptations in more detail.

*Structural brain changes [magnetic resonance imaging (MRI)].* Prolonged exposure to BR was shown to trigger significant structural changes in the brain, particularly within grey matter and white matter regions. Koppelmans et al. (2017) found that 70 days of HDTBR led to partial but not fully reversible volumetric changes in grey matter in sensorimotor brain regions, including the primary motor cortex, somatosensory cortex and cerebellum. These changes were associated with a decrease in functional mobility and balance, with grey matter changes in the precuneus cortex, precentral gyrus and postcentral gyrus correlating with a deterioration in balance. The study by Koppelmans et al. (2017) suggests that these findings reflect upward neuroplasticity and fluid shifts, analogous to the sensorimotor changes experienced by astronauts during spaceflight (Doroshin et al., 2022). Additionally, Roberts et al. (2015) identified structural shifts in brain morphology, including an upward displacement and posterior rotation of the brain within the skull after 90 days of HDTBR (the post-BR scans described were obtained between days BR42 and BR60). Increased tissue density in the frontoparietal lobes and decreased density along the base of the brain, including the orbitofrontal cortex, suggests that HDTBR-induced brain tissue redistribution could impact cerebrospinal fluid dynamics and intracranial pressure (Roberts et al., 2015). It should be noted that those data are exclusively derived from very long-term studies and future experiments need to investigate if these effects are comparable to short-term BR.

*Functional brain changes [functional MRI (fMRI)].* fMRI studies have extensively documented the brain's response to disuse analogues, revealing further insights into sensorimotor control adaptations. For example, Yuan, Koppelmans, Reuter-Lorenz, de Dios, Gadd, Riascos et al. (2018) reported increased activation in the cerebellum, fusiform gyrus, hippocampus and visual processing areas during foot movement tasks after 70 days of HDTBR. This increase in brain activity positively correlated with functional mobility and balance control, indicating adaptive changes in neural control during prolonged BR. Additionally, a greater increase in activation across multiple frontal, parietal and occipital regions during vestibular stimulation was associated with more significant declines in balance and mobility, suggesting reduced neural efficiency post HDTBR (Yuan, Koppelmans, Reuter-Lorenz, de Dios, Gadd, Wood et al., 2018). Despite the uniqueness of these results, interpretations should be made with caution because HDTBR was combined with exercises including resistance training, continuous aerobic exercise and high-intensity aerobic interval training in addition to routine stretching and physiotherapy (Koppelmans et al., 2017; Ploutz-Snyder

et al., 2014; Yuan, Koppelmans, Reuter-Lorenz, de Dios, Gadd, Riascos et al., 2018; Yuan, Koppelmans, Reuter-Lorenz, de Dios, Gadd, Wood et al., 2018).

In a short-term HDTBR of 3 days only, Liao et al. (2012) reported a reduction in resting-state connectivity in the left thalamus, suggesting an early functional disruption. Zhou et al. (2014) further reported reduced degree centrality in the left anterior insula and the anterior part of the middle cingulate cortex and detected that a functional network centred on these regions was significantly affected by 45 days of HDTBR. Furthermore, in a 70 day HDTBR study, Cassady et al. (2016) investigated the effects of mechanical unloading on resting-state brain connectivity and behaviour in 17 male participants. Significant changes were found with respect to functional connectivity in motor, somatosensory and vestibular brain areas that were associated with changes in sensorimotor and spatial working memory performance, suggesting that mechanisms of neuroplasticity may facilitate adaptation to the analogous environment of microgravity (Cassady et al., 2016). Finally, a study by McGregor et al. (2023) reported a group-specific change in connectivity between the posterior parietal cortex and somatosensory regions after 60 days of HDTBR. The HDTBR group showed increased connectivity, whereas the artificial gravity group showed decreased connectivity. In addition, participants who were exposed to HDTBR only and showed increased connectivity between the putamen and somatosensory cortex showed a greater decrease in mobility after HDTBR (McGregor et al., 2023).

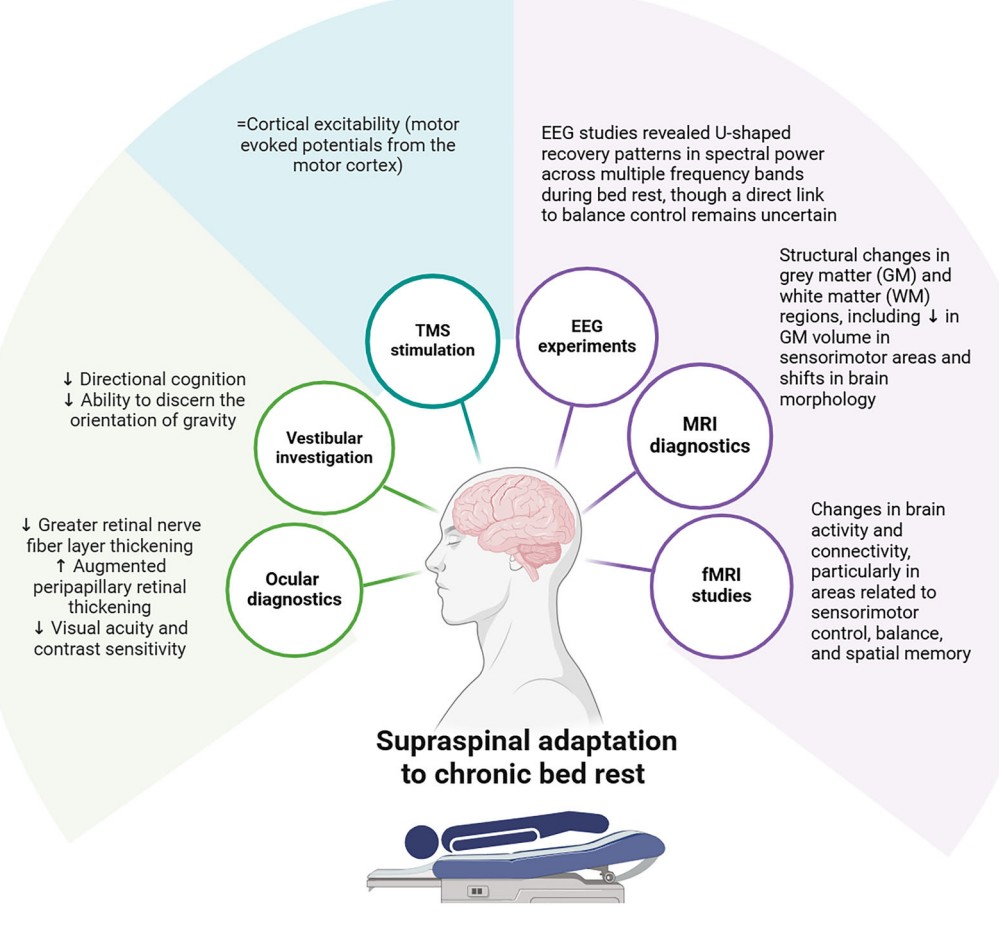

**Figure 2. Supraspinal adaptations across bed rest durations**
Methodological spectrum and primary results of all studies on supraspinal adaptation to chronic BR with a duration of 3–90 days. From left to right, sensory adaptations, results on cortical excitability, and structural and functional adaptations of different brain areas. The parameters presented were assessed at different durations of BR, although no clear time-dependent trend can be concluded.

*Brain electrocortical changes (EEG/event-related potentials).* EEG has only been used in a limited number of studies to assess the neurophysiological effects of prolonged BR. Initially, Han et al. (2001) found that short-term BR in simulated microgravity resulted in slowed peak frequencies and increased alpha, theta and beta1 activity, suggesting significant changes in the EEG power spectrum and possible effects on brain function. Further research indicates that HDTBR leads to significant changes in electrocortical activity, although these alterations tend to follow a U-shaped recovery trajectory, with initial reductions in EEG power that are later restored upon resumption of the upright position (Brauns et al., 2021). Brauns et al. (2021) also reported that, although overall functional connectivity remains stable during rest, there is a reduction in EEG spectral power across multiple frequency bands, including delta, theta, alpha and beta. Furthermore, EEG source localization revealed significantly lower activity in a broad range of centroparietal and occipital areas within the alpha and beta frequency domains (Brauns et al., 2021). These findings suggest a shift in cortical excitability that persists throughout HDTBR but normalizes relatively quickly after re-ambulation.

A review article by Marušič et al. (2014) highlighted that the most pronounced EEG changes during BR occur in the theta and alpha bands, suggesting cortical inhibition as a result of reduced sensory input. The same research group later performed analyses of event-related potentials before and after 14 days of horizontal BR and reported increased P100 and P200 amplitudes and decreased P100 latency in the BR-only group (Marusic, Pišot et al., 2021). Although these electrophysiological changes were not directly correlated with postural balance control, the same subjects showed reduced functional mobility after BR (Marusic et al., 2015), suggesting a possible link between altered cortical processing and impaired motor function.

One proposed mechanism for these EEG changes is the reduction in afferent and efferent nerve stimulation associated with prolonged HDTBR leading to reduced sensorimotor input. Marušič et al. (2014) reviewed (although only in the few available studies) that exposure to BR and HDTBR conditions leads to reduced cortical excitability, particularly in the theta and alpha frequency bands, probably reflecting reduced sensory feedback from the vestibular and proprioceptive systems, which are both critical for postural stability. Additionally, changes in cerebrospinal fluid distribution and brain shift may further contribute to these EEG alterations. Rice et al (2013) demonstrated that even minor changes in cerebrospinal fluid thickness as a result of postural shifts can significantly affect EEG signal magnitude, particularly in the occipital cortex. The interplay between reduced neural stimulation and fluid redistribution may play a key role in the neurophysiological adaptations to HDTBR and may contribute to the balance and mobility impairments observed after prolonged immobilization.

However, the direct relationship between electrocortical activity and postural control remains poorly understood. Future research should simultaneously assess EEG activity and postural control dynamics to better understand how neuronal adaptations during prolonged BR contribute to balance decline and how these changes are reversed upon re-ambulation. This integrative approach could provide critical insights into the neural mechanisms underlying postural instability and guide the development of targeted countermeasures for spaceflight, prolonged hospitalization, and other conditions associated with extended immobility.

**Neuronal plasticity and spinal motor control (proprio-spinal neurons/stretch reflex).** Neuronal plasticity on the spinal level has been assessed in several BR and HDTBR studies with an emphasis on the amplitudes and latencies of reflexes (Paloski et al., 2017). Reschke et al. (2009), Reschke et al. (2009), Sato et al. (2000) and Yamanaka et al. (1999) used a mixed model regression analysis and found an increased latency and amplitude of the monosynaptic stretch reflex of lower limb muscles in response to chronic 42–90 days of HDTBR persistent up to 8 days of recovery. Adaptations in latencies and amplitudes of monosynaptic reflexes were not related to postural degradation. However, the studies highlighted that dynamic head movements might affect posture after BR. The latency of the functional stretch reflex remained unchanged.

Paloski et al. (2017) assessed the amplitudes and latencies of the monosynaptic stretch reflex during and after 21 days of HDTBR. They found delayed latencies and decreased amplitudes of the monosynaptic stretch reflexes during HDTBR persistent beyond the period of immobility. Functional stretch reflexes, however, remained unaffected. After 21 days of HDTBR, the soleus H-reflex during standing was found to be reduced, without significant differences in motor-evoked potentials (MEPs) induced by transcranial magnetic stimulation (TMS) in the soleus muscle, suggesting a strong spinal inhibition of postural reflexes after BR induced disuse (Miyazaki et al., 2002). Yamanaka et al. (1999) applied peripheral nerve stimulation to elicit H-reflexes and elicit MEPs by TMS in the soleus muscle during quiet unsupported bipedal leg stance before and after 20 days of HDTBR. Amplitude of soleus H-reflexes and sizes normalized to the maximal M-wave during upright standing following HDTBR decreased in all subjects. MEPs remained unchanged before and after HDTBR. Moreover, MEP/$H_{max}$ ratios after HDTBR was larger than before in all subjects. These results indicate a strong and selective inhibition of H-reflex

and therefore a reduced spinal excitability as a result of long-term disuse. Such neuromotor adaptations may have particularly detrimental consequences for older adults who already exhibit age-related impairments in spinal reflex excitability and sensorimotor integration, compounding their risk of instability and falls. However, it must be stated that the neuromotor adaptations regarding the spinal excitability are not univocal among several disuse models. Increases $H_{max}/M_{max}$ ratios have been reported after short-term unilateral limb immobilization (Lundybse-Jensen & Nielsen, 2008) and after medium-term unilateral limb suspension (Seynnes et al., 2010). Similar results have been reported for a medium-term BR case study (Duchateau, 1995) and animal models (Islamov et al., 2011). The effect on balance control has not directly been assessed and inter-relationships have not been described. However, the changes indicate that the mono- and polysynaptic spinal reflexes, which play an important role in compensation postural instability, and the transmissibility at the alpha motor neuron are impaired by BR.

Finally, Manganotti et al. (2021) investigated peripheral nerve adaptation both motor and sensory functional characteristics and described a consistently decreased amplitude of F-waves from tibial posterior nerve and peroneal nerve known to respond to postural changes after 10 days of BR without changes of latency.

### Proprioception, vestibular and visual cues

*Proprioception.* Adaptation of sensorimotor interaction related to postural balance control in response to BR has been reported in terms of integration of sensory cues at spinal levels. On the spinal level, studies by Sato et al. (2000) and Yamanaka et al. (1999) refer to a reduced spinal excitability during and after several days of HDTBR suggesting an inhibition of Ia afferent muscle spindle input. Concomitantly, these studies demonstrated unison that latencies of peripheral reflexes were delayed during and after HDTBR (Haruna et al., 1999; Reschke & Paloski, 2017; Reschke et al., 2009; Yamanaka et al., 1999). Both adaptations are mostly accompanied by a reduced ability to counteract balance disturbances to regain postural stability because sensorimotor responses are simultaneously delayed and reduced in magnitude. Such alterations are further highlighted by significant reduced conduction velocities (Rescke et al., 2012) after 60–90 days of HDTBR of stretch reflexes that are paramount for fall avoidance after sudden postural disturbances. Muir et al. (2011) could not find any difference in neuro-sensitivity using two analytic approaches comparing baseline to 90 days of HDTBR.

*Vision.* Most experiments investigating visual acuity and reliance in postural stability compare eyes open and eyes closed paradigms in longitudinal study designs with periods of 14–70 days of BR and HDTBR (Clément et al., 2015; Haines, 1974; Kouzaki et al., 2007; Mulder et al., 2015; Paloski et al., 2017; Viguier et al., 2009). Some of these studies also combined the visual analytic diagnostics with either modifying the somatosensory and/or altering the vestibular input (Clément et al., 2015; Mulder et al., 2015; Paloski et al., 2017). The outcomes revealed that subjects demonstrate a strong vision effect on both sway trajectories in anterior/posterior plane and the medio/lateral plane after compared to before HDTBR for 60–70 day periods of chronic disuse (Viguier et al., 2009). By contrast, Koppelmans et al. (2015) and Tays et al. (2022) used the rod and frame test to measure reliance on visual vs. vestibular and proprioceptive cues after 60 or 70 days of HDTBR compared to before. Both studies have shown that visual reliance remained stable during BR indicating that the perception of the upright vertical orientation does not change during the time course of disuse.

The mechanistic explanation for the findings, especially for the contradictory results, is not yet based on solid experimental evidence, but there are at least three aspects related to ocular, cortical and functional cues that are crucial for understanding. Time-sensitive adaption of ocular structures and visual function retrieved from 14 to 21 days of HDTBR show greater retinal nerve fibre layer thickening and greater peripapillary retinal thickening, suggesting that time may affect the amount of nerval and optic disc swelling as a result of cephalad fluids shift in the sensory organ and afferent nerves (Gracheva et al., 2023; Taibbi et al., 2016). In addition to these structural adaptations, recent evidence indicates that BR also leads to a significant reduction in retinal venular diameter, suggesting alterations in microvascular regulation in response to prolonged inactivity (Saloň et al., 2023). These changes may reflect systemic vascular deconditioning, which could contribute to altered ocular perfusion and potentially affect visual function over extended periods of BR. Beyond the maladaptation of the sensory organ itself that allows humans to perceive visual information, 60–90 days of prolonged BR results in decreased visual acuity and contrast sensitivity, both of which are crucial for detecting environmental cues that aid postural stability (Reschke et al., 2009). Furthermore, from a brain processing perspective, chronic disuse leads to neural adaptations in the sensory cortex, diminishing its responsiveness to visual stimuli (Yuan, Koppelmans, Reuter-Lorenz, de Dios, Gadd, Riascos et al., 2018). These neural changes contribute to a decreased capacity to accurately process visual information, exacerbating balance impairments. This degenerated ocular function and compromised integration manifests as delayed or inappropriate postural responses when visual cues are ambiguous or insufficient, leading to instability and an increased risk of falls. The brain's diminished ability to adapt to changing visual

environments further exacerbates these balance issues, particularly in dynamic or low-visibility conditions.

*Vestibular system.* Vestibulo-spinal reflexes are intended to produce corrective sway responses in the event of an unintended loss of postural balance control, which makes them critical for human mobility and postural stability. Most studies on vestibular topics (Burgeat et al., 1981; Clément et al., 2015; Koppelmans et al., 2015, 2017; McGregor et al., 2023; Miller et al., 2018; Mulavara et al., 2018; Mulder et al., 2015; Yuan, Koppelmans, Reuter-Lorenz, de Dios, Gadd, Riascos et al., 2018) have assessed the sensitivity of postural balance control to vestibular conditions for BR and HDTBR periods of 14–70 days. Experimentally, head erect conditions predominantly with eyes closed were compared to head rotations in the pitch plane and modified auditory manipulation measured during the Sensory Organization Tests (i.e. SOT). The largest decrements were found with eyes closed compared to eyes open, on sway-referenced support and during head tilt (Clément et al., 2015; Paloski et al., 2017) and were interpreted as a reduced ability of the vestibular system to discern the orientation of gravity after chronic BR. Compromised vestibular function as a result of long-term disuse has been mechanistically explained by attenuated speed of vestibular-ocular reflexes provoked by nystagmus after 7 days of BR (Burgeat et al., 1981) and 60 days of HDTBR (Mackenzie et al., 2024) and by increased directional variability and the associated sway path elicited by stochastic vestibular stimulation after 60 days of BR still being elevated until 6 days of recovery. Decremented directional cognition is of critical relevance because the vestibular organ provides the signals to the vestibular cortex that governs the sensorimotor transformation process in which head-centred sensory information is converted into accurate body-centred, 360° directional balance reactions discerning the direction of gravity (Clément et al., 2015). Experimentally demonstrated by vestibular stimulation and fMRI, the BR-induced decrement is compensated by gradually increased activation of the bilateral insular cortex, suggesting that vestibular inputs might be upregulated as a result of the reduced somatosensory inputs experienced during the 70 days of HDTBR (Yuan, Koppelmans, Reuter-Lorenz, de Dios, Gadd, Riascos et al., 2018). Furthermore, the insula, frontal and parietal areas exhibited cumulative increases in activation for vestibular stimulation during the course of 70 days of HDTBR and greater increases were associated with larger declines in locomotor and postural balance control after BR, suggesting that more neural resources were required to process vestibular input.

**Structural adaptation in the brain and spine in animal models.** Different species, from quadrupeds to humans, actively maintain a basic, upright or dorsal side-up body posture (Maus et al., 2010). In particular, animal experiments provide information about structural adaptations in the brain and spinal cord that cannot be investigated in humans. Comparative studies of neuronal mechanisms are based on the assumption that the nervous control of anti-gravity behaviour has similar solutions in different species, and thus the results obtained using animal models on adaptations to bed rest and immobilization may have significance for understanding the mechanisms of postural balance control in humans (Maus et al., 2010).

Maintenance of this posture is a non-volitional activity that is based, in many species, on innate neural mechanisms. Adaptations to immobilization showed that spinal neural circuitries are subject to changes as a result of immobilization reduction in the excitability of spinal motor neurons and changes in synaptic plasticity, including modifications to both long-term potentiation and long-term depression processes. A 1–6 week immobilization of the knee joint in guinea pigs revealed retrograde degeneration of motor neurons in the spinal cord including demyelination of axons, reduction of cell organelles, indentation of the nuclear envelop and nuclei clumping (He & Dishman, 2010). Interestingly, a study by Langlet et al. (2012) confirmed these results in rats and found that 14 days of hindlimb unloading led to a dramatic reduction (–61%) of hindlimb representation on the M1 cortex. These changes in spinal cord structure and plasticity may contribute to reduced muscle force generation and impairments in reflexive motor control negatively impacting the control of stable posture.

Adaptive changes in the motor cortex involved in voluntary movement during and after forelimb immobilization in adult rats (Viaro et al., 2014) induced bilateral cortical hypo-excitability that progressively decreased over 30 days of immobilization and, after cast removal, steadily increased, but remained partial at 15 days. Furthermore, impairment of intracortical synaptic connectivity has been documented. In young rats, histological alterations in the sensory cortex occurred concomitant with impaired achievement of developmental milestones and motor skills (Marcuzzo et al., 2010).

Although animal models provide valuable insights into the structural and functional changes associated with BR, care must be taken when directly extrapolating these findings to human physiology. Differences in neuroanatomy, sensory integration and motor control across species may lead to variations in the extent and nature of neuromotor adaptations. However, the underlying mechanisms observed in animal studies, such as neural plasticity and sensory processing alterations, offer important parallels that can inform our understanding

of postural control changes in humans during periods of disuse.

## Effect of the selected models of disuse: horizontal and head-down tilt bed rest

When comparing outcomes between standard horizontal BR and 6–8° HDTBR, potential differences in their effects on postural balance control and balance are difficult to distinguish. The reason for this is limited evidence because only very few studies have been published on horizontal BR (Haines, 1974; Manganotti et al., 2021; Morishima et al., 1997; Reschke et al., 2009; Sarabon & Rosker, 2013; Šarabon et al., 2018) compared to more than 50 articles related to HDTBR. Standard horizontal BR primarily induces musculoskeletal deconditioning as a result of inactivity, including muscle atrophy, reduced proprioceptive feedback, and a decline in overall balance control and postural stability upon re-ambulation. However, it is speculated, without experimental evidence, that the vestibular system remains relatively unaffected, which may lead to less severe impairments in sensory integration for balance.

By contrast, 4–8° HDTBR introduces additional physiological stress by simulating a microgravity environment, including fluid shifts towards the head (Barkaszi et al., 2022; Koppelmans et al., 2017) and altered vestibular input (Mackenzie et al., 2024; Miller et al., 2018; Mulder et al., 2015; Yuan, Koppelmans, Reuter-Lorenz, de Dios, Gadd, Riascos et al., 2018). These changes may lead to more profound disruptions in the integration of sensory information necessary for maintaining postural balance control. HDTBR may therefore exacerbate rebalancing challenges (Koppelmans et al., 2017; Miller et al., 2018; Muir et al., 2011), where the brain must recalibrate the integration of visual, vestibular and proprioceptive inputs (Barkaszi et al., 2022; Koppelmans et al., 2017). This makes HDTBR a more rigorous model for studying the complex interactions between sensory systems under conditions similar to microgravity, such as during spaceflight.

## Delayed restorations of neuronal deficits after disuse

Chronic disuse leads to significant impairments in postural balance and motor control, and the origin of these deficits is attributed to disturbances in neural drive involving the peripheral and central nervous system. Static and dynamic postural instability has been reported to persists beyond prologued BR and HDTBR for 7 days (Burgeat et al., 1981), 2 weeks (Ritzmann et al., 2018; Sarabon & Rosker, 2015) up to 4 weeks (Dupui et al. 1992) after re-ambulation. Similarly, recovery of spinal reflex latencies and amplitudes, as well as muscle activation patterns, was reported to be gradual, with significant improvements observed within the first few days of re-ambulation (Paloski et al., 2017; Ritzmann et al., 2018). For example, after 21 days of HDTBR, the amplitude of the monosynaptic stretch reflex returned to baseline within 8 days of BR, indicating a rapid recovery of spinal reflex pathways (Paloski et al., 2017). The recovery of antagonistic muscle activation after 60 days of HDTBR took 2 weeks (Ritzmann et al., 2018). No follow-up studies exist on the brain adaptation. However, persistency for almost 1 week has been shown for visual (Ritzmann et al., 2018) and vestibular cues (Mackenzie et al., 2024), which translates towards the involvement supraspinal areas such as the cerebellum or the sensor corticis (McGregor et al., 2023). Caution is advised during this sensitive 1–4 week phase after re-ambulation because clinically relevant fragility exposes the individual to a greatly increased risk of injury in daily life (Mulder et al., 2015; Stuempfle & Drury, 2007).

## Knowledge gaps

Despite advances in understanding the effects of BR on postural balance control and neuromotor function, several critical gaps in our knowledge remain. Today, there is insufficient mechanistic understanding of how sensory impact, particularly of the vestibular (Gracheva et al., 2023; Taibbi et al., 2016) and visual systems (Clément et al., 2015; Yuan, Koppelmans, Reuter-Lorenz, de Dios, Gadd, Riascos et al., 2018), deteriorates with prolonged BR and whether proprioceptors such as muscle spindles and Golgi tendon organs adapt in a similar manner (Saumur et al., 2020). In addition, the influence of sex and age on these discrepancies has not yet been sufficiently researched, with women and older adults being less researched but more probably at increased risk as a result of endocrine prerequisites and the physiological signs of senility (Deschenes et al., 2008; Pišot et al., 2016). The different effects of short-, medium- and long-term BR (Marusic, Narici et al., 2021) also need to be investigated further, as well as the effects on the time to regain full postural capacity within 1 month after the BR intervention and the optimization of measurement protocols to avoid participant fatigue (Marusic, Narici et al., 2021; Pišot et al., 2016). Furthermore, methodologically advanced imaging techniques such as fMRI, brain connectivity with mathematical modelling or even the measurement of brain dynamics during postural balance control and mobility tasks with approaches such as mobile brain/body imaging (Gramann et al., 2011; Marusic et al., 2023) are not yet sufficiently utilized in BR studies, although they have the potential to provide deeper insights into brain changes during disuse. Lastly, assessing both static and dynamic postural control following bed-rest protocols is crucial for a comprehensive understanding of the effects of physical deconditioning on postural stability. Because

both components appear to differ in their underlying determinants (Paillard, 2017), BR may differentially affect static and dynamic postural balance control.

## Conclusions and future directions

This review synthesizes and resonates the impact of BR on brain function and structure, spinal circuitries, and proprioception, vestibular and visual cues, which are essential for maintaining postural balance (Fig. 2). The persistence of these brain changes throughout BR and their return to baseline upon resumption of the upright position involving the entire brain in regard to function and the primary motor cortex, somatosensory cortex and cerebellum in regard to morphology suggest a direct link between BR and alterations in brain structure and function. The reduction in spinal excitability and the delayed sensorimotor responses during and after BR are particularly concerning because they lead to a decreased ability to counteract balance disturbances. Visual acuity and contrast sensitivity, which are crucial for detecting environmental cues, are also negatively affected by prolonged BR, further compromising postural stability. The findings provide a thorough examination of the neuromotor changes in postural balance control following BR and underscore the complexity of these changes, involving not only muscle strength, but also significant alterations in neural control and sensory processing. These insights are critical for developing targeted interventions aiming to mitigate the adverse effects of prolonged immobility on postural balance control.

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

## Additional information

### Competing interests

The authors declare that they have no competing interests.

### Author contributions

All authors contributed to the conception or design of the work. R.R., J.W. and U.M. drafted the work, and all authors revised it critically for important intellectual content. All authors approved the final version of the manuscript submitted for publication.

### Funding

This work was supported by European Commission (EC): Uros Marusic, 101120150.

### Acknowledgements

### Keywords

atrophy, balance, deconditioning, disuse, immobilization, muscle, motor evoked potential, posture, proprioception, reflexes, sensory, sensorimotor, upright stance, vestibular, visual

### Supporting information

Additional supporting information can be found online in the Supporting Information section at the end of the HTML view of the article. Supporting information files available:

**Peer Review History**

