## [Peer Review History · The Journal of Physiology]

Neuromotor changes in postural control following bed rest

Uros Marusic, Christoph Centner, Luke Hughes, Janice Waldvogel, and Ramona Ritzmann

DOI: 10.1113/JP285668

Corresponding author(s): Ramona Ritzmann (ramona.ritzmann@sport.uni-freiburg.de)

Review Timeline:

Submission Date:	31-Aug-2024
Editorial Decision:	11-Nov-2024
Revision Received:	28-Feb-2025
Accepted:	05-Mar-2025

Senior Editor: Richard Carson

Reviewing Editor: Matthew Fogarty

Transaction Report:

Dear Professor Marusic,

Re: JP-TR-2024-285668 "Neuromuscular changes in postural control following bed rest" by Uros Marusic, Christoph Centner, Luke Hughes, Janice Waldvogel, and Ramona Ritzmann

Thank you for submitting your manuscript to The Journal of Physiology. It has been assessed by a Reviewing Editor and by 1 expert referee and we are pleased to tell you that it is acceptable for publication following satisfactory revision.

ABSTRACT FIGURES: Authors may use The Journal's premium BioRender account to create/redraw their Abstract Figures (and any other suitable schematic figure). Information on how to access this account is here: <https://physoc.onlinelibrary.wiley.com/journal/14697793/biorender-access>.

REVISION CHECKLIST: Upload a full Response to Referees file. To create your 'Response to Referees' copy all the reports, including any comments from the Senior and Reviewing Editors, into a Microsoft Word, or similar, file and respond to each point, using font or background colour to distinguish comments and responses and upload as the required file type.

We look forward to receiving your revised submission.

Yours sincerely,

Richard Carson
Senior Editor

EDITOR COMMENTS

Reviewing Editor:

I thank the authors for their highly readable review of the neuromotor changes in balance following bed rest.

I have a few specific comments for the authors.

1 - I think in the title and the manuscript "neuromuscular" should be "neuromotor". Neuromuscular has a specific connotation for nerve-muscle interactions and so on, whereas I think the focus here is movement, rather than NMJs per se.

2 - Some context would be good for the intro - how short is acute, how long is chronic months? weeks?

3 - The focus seems to be spaceflight, which is applicable to a very small proportion of society. I think a major point of reference is lost in the manuscript by not bringing up age. The elderly are more at risk of falls, have comorbidities that predispose to extended hospitalisation and may exhibit sarcopaenia/frailty. Some attention to this issue would greatly enhance the potential readership.

4 - Are these studies outlined done in healthy volunteers? what conditions cause bed rest? Is there an additive effect of age (see above)?

5 - Perhaps a small 1-2 paragraph section on mechanistic work in animal models may round this out very nicely.

Senior Editor:

As noted in comments provided by the referee and Reviewing Editor, it is critical that it can be shown that the inferences drawn do not depend on specific populations or specific experimental models. Given the broad audience of physiologists represented by readers of the Journal, I strongly recommend that the authors expand their presentation (this may require additional content) with a view to demonstrating the generality of the adaptive processes under consideration.

REFeree COMMENTS

Referee #1:

This narrative review explores the impairment of postural balance control following experimental bed rest and the potential mechanisms driving this maladaptation. To the best of my knowledge, this is the first review on this specific topic. The subject is interdisciplinary, of significant physiological relevance, with clear translational value to both clinical settings (e.g. bed rest due to illness and injury) and space-related contexts, such as in astronaut health. It is commendable that the Authors not only reviewed the existing evidence but also provided novel perspectives, such as the comparison between horizontal bed rest and head-down tilt bed rest (Section 4) and insights into the recovery of postural balance control (Section 5). Overall, this is an interesting and well-written review article; however, several points require attention for further improvement:

1) While the title claims that "neuromuscular changes" are reviewed, the article exclusively focuses on neural factors (supraspinal and spinal level) and sensory feedback integration. Although I agree that these are the most critical aspects involved in postural balance control maintenance, the potential contribution of the "muscular" component is not sufficiently addressed. The Authors mentioned a single study suggesting muscle role might be minor (L75-76), but this view is somewhat simplistic. Muscle strength might be more relevant in older adults and dynamic conditions (Paillard, 2017; see

also Point 2), and tendon properties may also play a role (Onambele et al., 2006). Moreover, I would expect the peripheral nervous system, which tunes information from the central nervous system, to be also likely involved in this process. Anthropometry (e.g. changes in body weight induced by bed rest) (Chiari et al., 2002; Hue et al., 2007) may also influence the biomechanics of postural balance control. I suggest that the authors either briefly discuss these aspects in dedicated paragraphs or more clearly acknowledge and justify their specific focus on neural and sensory factors, adjusting the title and text accordingly.

2) It is now well-established in the literature that static and dynamic postural balance control are not interchangeable and represent well-distinct paradigms (Ringhof & Stein, 2018; Rizzato et al., 2021). For instance, previous studies suggested employing dynamic rather than static tests to detect postural balance control impairments resulting from previous injury (Ross & Guskiwicz, 2004) or to study the impact of acute physical exercise on postural balance control (Marcolin et al., 2019). Moreover, the determinants of static and dynamic postural balance control potentially differ (Paillard, 2017). As the Authors mentioned both static and dynamic balance were assessed in previous bed rest studies (L113), this distinction should be explored better throughout the review.

3) In some paragraphs (e.g., 3.1.3; 3.2), the authors should not only describe the specific physiological alterations occurring with bed rest but also explain how these alterations could affect postural balance control in this context. See also my comment on sleep behaviour in the 'Specific Comments' section.

4) Based on the literature, is it possible to propose a potential 'time course' for these alterations during bed rest? Among the different proposed mechanisms, which occur earlier and which develop at a later stage?

5) Adding a table summarising the designs and findings of the studies described in Section 2 would improve accessibility for readers by providing a detailed overview of the investigations on postural balance control alterations during bed rest. The table should include elements such as study design, sample size, methods employed, and the type of balance assessed, offering a clearer snapshot of the relevant research, as this is not described in detail in the text.

6) The Authors used interchangeably the terms "balance control", "postural equilibrium", "postural control", and "balance". I recommend the Authors to keep the terminology consistent throughout the manuscript. To the best of my knowledge, the term "postural balance control" is considered the most complete and is generally employed by experts in the field (E.g. Marcolin et al., 2022).

Specific Comments:

Title page: Author order and corresponding author in the manuscript do not coincide with the submission information on the website. Please clarify this aspect.

L118: Is there any study exploring the effects of dual-task on postural balance control following bed rest? If yes, this should be highlighted in this paragraph.

L143: Specify that the higher prevalence of falls was tested experimentally and not observed in a "real world" scenario.

L167-182: Observations from this paragraph are exclusively derived from very long-term bed rest studies. The Authors are invited to acknowledge that results may differ with short-term bed rest.

L240-249: Are the changes in EEG patterns during sleep relevant to postural balance control? If so, the authors should explain the connection. If not, I suggest removing this paragraph to maintain focus.

L276-277: To the best of my knowledge, the opposite trend was often observed in the disuse literature. A previous bed rest case study found an increased Hmax/Mmax ratio (Duchateau, 1995). Similar findings were also reported in other disuse models in humans (Lundbye-Jensen & Nielsen, 2008; Seynnes et al., 2010), and in related animal studies (De-Doncker et al., 2006; Islamov et al., 2011). I understand this review is not the place for a methodological discussion on the topic, but I suggest the Authors consider reporting these findings are not univocal.

L301: The Authors may consider also adding to this paragraph this recent manuscript on changes in retinal venular vessel diameters: (Saloň et al., 2023).

L365: Have the Authors considered expanding this paragraph with a comparison of bed rest with other disuse models, such as unilateral lower limb suspension or immobilisation? While these unilateral models can also disrupt postural balance control (Elam et al., 2022), the mechanisms may differ due to the lack of fluid shift and systemic effects, as well as the lower likelihood of impacting vision and causing structural brain changes. While I do not consider this a requirement, as the authors have motivated their focus on bed rest, it could add an interesting dimension to the manuscript.

L450: The author contribution section seems more referred to an original research article rather than a narrative review. Please amend it accordingly.

L712 (Figure 2): In order to enhance the Figure's readability and make it more schematic, I recommend the Authors change, where possible, verbs with symbols. E.g. "↓ directional cognition" instead of "decreased directional cognition".

L712 (Figure 2): Are there some parameters changing with long but not short-term bed rest? If yes, the Authors are invited to specify this in the figure or figure caption.

References:

- Chiari L, Rocchi L & Cappello A (2002). Stabilometric parameters are affected by anthropometry and foot placement. *Clin Biomech* 17, 666-677.
- De-Doncker L, Kasri M & Falempin M (2006). Soleus motoneuron excitability after rat hindlimb unloading using histology and a new electrophysiological approach to record a neurographic analogue of the H-reflex. *Exp Neurol* 201, 368-374.
- Duchateau J (1995). Bed rest induces neural and contractile adaptations in triceps surae. *Med Sci Sports Exerc* 27, 1581-1589.
- Elam C, Hvid LG, Christensen U, Kjær M, Magnusson SP, Aagaard P, Kall LB & Suetta C (2022). Effects Of Aging On Changes In Muscle Power And Postural Control After Immobilization And Re-training. *J Musculoskelet Neuronal Interact* 22, 486-197.
- Hue O, Simoneau M, Marcotte J, Berrigan F, Doré J, Marceau P, Marceau S, Tremblay A & Teasdale N (2007). Body weight is a strong predictor of postural stability. *Gait Posture* 26, 32-38.
- Islamov RR, Mishagina EA, Tyapkina O V., Shajmardanova GF, Eremeev AA, Kozlovskaya IB, Nikolskij EE & Grigorjev AI (2011). Mechanisms of spinal motoneurons survival in rats under simulated hypogravity on earth. *Acta Astronaut* 68, 1469-1477.
- Lundbye-Jensen J & Nielsen JB (2008). Immobilization induces changes in presynaptic control of group Ia afferents in healthy humans. *J Physiol* 586, 4121-4135.
- Marcolin G, Panizzolo FA, Biancato E, Cognolato M, Petrone N & Paoli A (2019). Moderate treadmill run worsened static but not dynamic postural stability of healthy individuals. *Eur J Appl Physiol* 119, 841-846.
- Marcolin G, Supej M & Paillard T (2022). Editorial : Postural Balance Control in Sport and Exercise. *Front Physiology* 13, 12-13.
- Onambele GL, Narici M V. & Maganaris CN (2006). Calf muscle-tendon properties and postural balance in old age. *J Appl Physiol* 100, 2048-2056.
- Paillard T (2017). Relationship between Muscle Function, Muscle Typology and Postural Performance According to Different Postural Conditions in Young and Older Adults. *Front Physiol* 8, 1-6.
- Ringhof S & Stein T (2018). Biomechanical assessment of dynamic balance: Specificity of different balance tests. *Hum Mov Sci* 58, 140-147.
- Rizzato A, Paoli A, Andretta M, Vidorin F & Marcolin G (2021). Are Static and Dynamic Postural Balance Assessments Two Sides of the Same Coin? A Cross-Sectional Study in the Older Adults. *Front Physiol* 12, 1-8.
- Ross SE & Guskiewicz KM (2004). Examination of static and dynamic postural stability in individuals with functionally stable and unstable ankles. *Clin J Sport Med* 14, 332-338.
- Saloň A, Çiftci GM, Zubac D, Šimunič B, Pišot R, Narici M, Fredriksen PM, Ngwenchi B, Chungag N, Sourij H, Šerý O, Zalaudek KS & Steuber B (2023). Retinal venular vessel diameters are smaller during ten days of bed rest. *Sci Rep* 1-7.
- Seynnes OR, Maffiuletti NA, Horstman AM & Narici M V. (2010). Increased H-reflex excitability is not accompanied by changes in neural drive following 24 days of unilateral lower limb suspension. *Muscle Nerve* 42, 749-755.

REQUIRED ITEMS

- Please include an Abstract Figure file, as well as the Figure Legend text within the main article file. The Abstract Figure is a piece of artwork designed to give readers an immediate understanding of the Review Article and should summarise the main conclusions. If possible, the image should be easily 'readable' from left to right or top to bottom. It should show the physiological relevance of the Review so readers can assess the importance and content of the article. Abstract Figures should not merely recapitulate other figures in the Review. Please try to keep the diagram as simple as possible and without superfluous information that may distract from the main conclusion of the Review. Abstract Figures must be provided by authors no later than the revised manuscript stage and should be uploaded as a separate file during online submission labelled as File Type 'Abstract Figure'. Please ensure that you include the figure legend in the main article file. All Abstract Figures will be sent to a professional illustrator for redrawing and you may be asked to approve the redrawn figure before your paper is accepted.

- Please upload separate high quality figure files via the submission form.

- Author profile(s) must be uploaded via the submission form. Authors should submit a short biography (no more than 100 words for one author or 150 words in total for two authors) and a portrait photograph of the two leading authors on the paper. These should be uploaded and clearly labelled together in a Word document with the revised version of the manuscript. Any standard image format for the photograph is acceptable, but the resolution should be at least 300 DPI and preferably more. A group photograph of all authors is also acceptable, providing the biography for the whole group does not exceed 150 words.

END OF COMMENTS

Dear Professor Marusic,

Re: JP-TR-2024-285668 "Neuromuscular changes in postural control following bed rest" by Uros Marusic, Christoph Centner, Luke Hughes, Janice Waldvogel, and Ramona Ritzmann

Thank you for submitting your manuscript to The Journal of Physiology. It has been assessed by a Reviewing Editor and by 1 expert referee and we are pleased to tell you that it is acceptable for publication following satisfactory revision.

Your revised manuscript should be submitted online using the link in your Author Tasks: <https://jp.msubmit.net/cgi-bin/main.plex?el=A2JS6FyI5A7ZLb3F7A9ftdyNG6i8LMCsroqnKVdPtvQZ>. This link is accessible via your account as Corresponding Author; it is not available to your co-authors. If this presents a problem, please contact journal staff (jp@physoc.org). Image files from the previous version are retained on the system. Please ensure you replace or remove any files that are being revised.

ABSTRACT FIGURES: Authors may use The Journal's premium BioRender account to create/redraw their Abstract Figures (and any other suitable schematic figure). Information on how to access this account is here: <https://physoc.onlinelibrary.wiley.com/journal/14697793/biorender-access>.

LANGUAGE EDITING AND SUPPORT FOR PUBLICATION: If you would like help with English

language editing, or other article preparation support, Wiley Editing Services offers expert help, including English Language Editing, as well as translation, manuscript formatting, and figure formatting at www.wileyauthors.com/eoo/preparation. You can also find resources for Preparing Your Article for general guidance about writing and preparing your manuscript at www.wileyauthors.com/eoo/prepresources.

REVISION CHECKLIST: Upload a full Response to Referees file. To create your 'Response to Referees' copy all the reports, including any comments from the Senior and Reviewing Editors, into a Microsoft Word, or similar, file and respond to each point, using font or background colour to distinguish comments and responses and upload as the required file type.

We look forward to receiving your revised submission.

Yours sincerely,

Richard Carson
Senior Editor
The Journal of Physiology

EDITOR COMMENTS

Reviewing Editor:

I thank the authors for their highly readable review of the neuromotor changes in balance following bed rest.

I have a few specific comments for the authors.

1 - I think in the title and the manuscript "neuromuscular" should be "neuromotor". Neuromuscular has a specific connotation for nerve-muscle interactions and so on, whereas I think the focus here is movement, rather than NMJs per se.

Reply: *We thank the reviewing editor for this important comment. We agree that "neuromotor" indeed fits better what we describe in our manuscript. "Neuromuscular" as replaced with "neuromotor" throughout the entire manuscript.*

2 - Some context would be good for the intro - how short is acute, how long is chronic months? weeks?

Reply: *Thank you for mentioning this point. We have added more information in the introduction. Based on previous systematic reviews (<https://journals.physiology.org/doi/pdf/10.1152/jappphysiol.00363.2020>), we defined short-term/acute as < 2 weeks, medium-term as 3-5 weeks and long-term as ≥ 6 weeks)*

3 - The focus seems to be spaceflight, which is applicable to a very small proportion of society. I think a major point of reference is lost in the manuscript by not bringing up age. The elderly are more at risk of falls, have comorbidities that predispose to extended hospitalisation and may exhibit sarcopaenia/frailty. Some attention to this issue would greatly enhance the potential readership.

Reply: *We appreciate the reviewer's insightful comment regarding the impact of age on bed rest outcomes. To address this, we have added specific text to the increased vulnerability of older adults due to sarcopenia, frailty, and impaired neuromotor function. Please see rows 69-71; 84-86.*

4 - Are these studies outlined done in healthy volunteers? what conditions case bed rest? Is there an additive effect of age (see above)?

Reply: *We appreciate the reviewer's thoughtful comment regarding the use of only healthy volunteers in the study and the implications this might have for understanding the effects of long-term bed rest in different populations, particularly those who are diseased or aged.*

1. **Justification for Healthy Volunteers:**

The primary rationale for using only healthy volunteers in this study was to establish a baseline understanding of neuromotor changes in balance control due to prolonged bed rest in an otherwise normal physiological state. Healthy participants allow us to isolate the specific effects of bed rest without the confounding influence of pre-existing medical conditions or age-related impairments. By starting with a "healthy" cohort, we aim to better understand the fundamental mechanisms at play in neuromotor control that can later be generalized to other populations with proper adjustments.

2. **Impact on Diseased Populations:**

In individuals with existing medical conditions (e.g., neurological diseases such as Parkinson's or Multiple Sclerosis, cardiovascular disease, etc.), the neuromotor system may already be compromised, which could alter the response to bed rest. For instance, patients with neurological conditions may exhibit greater difficulties in motor coordination, postural control, and balance even before bed rest. Long-term bed rest could exacerbate these impairments, leading to more pronounced disruptions in motor function, postural control, and sensory-motor integration, particularly if the condition involves muscle weakness, reduced proprioception, or compromised central nervous system regulation of balance.

It is important to consider that these populations might also have slower recovery times or may experience a more pronounced deconditioning effect compared to healthy individuals. Moreover, the mechanisms underlying these impairments could differ from those observed in healthy subjects, making it critical to consider their specific pathophysiology when interpreting the effects of bed rest.

3. **Impact on Aged Populations:**

In the elderly, aging itself is associated with a decline in neuromotor control, including reduced muscle mass and strength (sarcopenia), changes in the vestibular system, decreased proprioception, and slower neural processing. These age-related changes may make older adults more vulnerable to the effects of bed rest, potentially leading to a more rapid decline in balance control and postural stability. The cumulative effect of aging-related changes in muscle and sensory systems could also lead to greater difficulty in recovering balance post-bed rest. This population may also experience greater risk of falls and longer recovery periods compared to younger or healthy controls.

4. **Differences in Responses to Bed Rest:**

Based on the differences in baseline neuromotor health and resilience, we hypothesize that diseased and aged populations may experience a more severe deterioration in balance control during and after bed rest. However, it is important to recognize that the underlying mechanisms in these populations may differ. For instance, in aged individuals, the decline in neuromotor function may stem primarily from muscle atrophy and reduced central nervous system efficiency, whereas in disease states, it could be due to a combination of factors such as inflammation, reduced neural plasticity, or altered muscle activation patterns.

Future studies would need to specifically address these questions by including diseased and aged populations to explore how pre-existing conditions or age-related changes influence the response to prolonged inactivity. We acknowledge that this is a limitation of the current study and we agree that future work in these areas is

necessary to gain a more comprehensive understanding of the full range of potential effects.

The differences in how diseased or aged individuals would respond to bed rest highlight the need for tailored interventions and rehabilitation strategies. For example, older adults or patients with certain diseases might require different types or intensities of physical therapy during or after bed rest to mitigate the decline in neuromotor function. Furthermore, it would be essential to monitor potential differences in recovery times, and to account for the potential for compensatory mechanisms that might be less effective in these populations.

5 - Perhaps a small 1-2 paragraph section on mechanistic work in animal models may round this out very nicely.

Reply: *We agree that including insights from mechanistic work in animal models would enhance the overall depth of the manuscript. Animals and humans actively stabilize a definite body orientation in space due to activity of the postural system. Animal models are invaluable for understanding the neurophysiological mechanisms that underlie (sensori)motor changes in balance control during and after prolonged bed rest, as they allow for more controlled manipulations and detailed observations of specific pathways involved in balance control.*

Studies in animal models, particularly rodent models, have provided significant insights into the impact of immobilization and bed rest on neuromotor function. However, none of the studies investigated the subsequent validity of chain of reasoning including the effect on the postural control of animals and its transferability to humans, as animals, unlike humans, stand on 4 legs. The biomechanical support surface is therefore larger and the anterior-posterior and medio-lateral stability is significantly greater in quadrupeds than in bipedal humans. Furthermore, the spinal column is not used in a vertical axis as in humans, but predominantly in a horizontal axis or in variably-changing axes for specific movement dynamics. As illustrated in the article of Donald et al. (2008) segmental kinematics, movement patterns and motor strategies are completely different in animals than in humans. Further, the neuronal requirements and command of skeletal muscles to keep stable upright posture operates different between humans and animals. From a mechanistic point of when it comes to bed rest, unloading conditions move in the center of interest, where the study of Duysens et al. (2000) provides valuable insight into supraspinal and spinal mechanisms in balance and gait control.

Research has demonstrated that prolonged inactivity can lead to changes in the musculoskeletal system (e.g., muscle atrophy and altered muscle fiber composition), as well as in neural circuits responsible for balance control. Animal studies have shown that long-term unloading (i.e., hindlimb suspension in rodents) results in altered proprioceptive feedback and reduced neuromuscular coordination. Both are key factors in impaired balance control. Additionally, these models have been instrumental in studying the effects of deconditioning on the central nervous system, revealing changes in motor cortex activity, cerebellar function, and even vestibular system plasticity following prolonged periods of inactivity.

While there are clear differences in how these mechanisms may manifest in humans versus animals, the findings from these studies are crucial for informing our understanding of how similar changes may occur in humans during long-term bed rest, particularly in populations at higher risk of deconditioning, such as the elderly or those with pre-existing medical conditions. Therefore, we added paragraph 3.4.

Donald C. Dunbar, Jane M. Macpherson, Roger W. Simmons, Athina Zarcades; Stabilization and mobility of the head, neck and trunk in horses during overground locomotion: comparisons with humans and other primates. J Exp Biol 15 December 2008; 211 (24): 3889–3907. doi: <https://doi.org/10.1242/jeb.020578>

Duysens J., Clarac F., Cruse H. (2000) Load-Regulating Mechanisms in Gait and Posture: Comparative Aspects. 01 JAN 2000 <https://doi.org/10.1152/physrev.2000.80.1.83>

Senior Editor:

As noted in comments provided by the referee and Reviewing Editor, it is critical that it can be shown that the inferences drawn do not depend on specific populations or specific experimental models. Given the broad audience of physiologists represented by readers of the Journal, I strongly recommend that the authors expand their presentation (this may require additional content) with a view to demonstrating the generality of the adaptive processes under consideration.

***Reply:** We appreciate the Senior Editor's recommendation to strengthen the generalizability of our discussion on adaptive processes. In response, we have expanded the manuscript to further contextualize the applicability of bed rest studies beyond specific populations and experimental models.*

Firstly, age and health status considerations: we have explicitly addressed the role of aging and pre-existing medical conditions by integrating new content on how older adults and individuals with neurological or cardiovascular conditions may experience more pronounced and potentially irreversible neuromotor impairments following bed rest.

Secondly, mechanistic Insights from animal models: we have introduced a dedicated section (3.4) discussing findings from animal studies that offer mechanistic insights into neuromotor adaptations to unloading conditions, emphasizing both the parallels and limitations of translating such results to human physiology. This addition ensures a broader scope of interpretation relevant to a diverse readership.

Finally, justification for study populations: we have clarified why healthy volunteers are commonly used in bed rest studies, emphasizing that these studies provide a controlled baseline for understanding neuromotor adaptation, which can then be extrapolated to clinical and aging populations with necessary modifications. We also highlight the need for future studies targeting diseased and elderly individuals to explore population-specific neuromotor decline and recovery dynamics.

REFEREE COMMENTS

Referee #1:

This narrative review explores the impairment of postural balance control following experimental bed rest and the potential mechanisms driving this maladaptation. To the best of my knowledge, this is the first review on this specific topic. The subject is interdisciplinary, of significant physiological relevance, with clear translational value to both clinical settings (e.g. bed rest due to illness and injury) and space-related contexts, such as in astronaut health. It is commendable that the Authors not only reviewed the existing evidence but also provided novel perspectives, such as the comparison between horizontal bed rest and head-down tilt bed rest (Section 4) and insights into the recovery of postural balance control (Section 5). Overall, this is an interesting and well-written review article; however, several

points require attention for further improvement:

1) While the title claims that "neuromuscular changes" are reviewed, the article exclusively focuses on neural factors (supraspinal and spinal level) and sensory feedback integration. Although I agree that these are the most critical aspects involved in postural balance control maintenance, the potential contribution of the "muscular" component is not sufficiently addressed. The Authors mentioned a single study suggesting muscle role might be minor (L75-76), but this view is somewhat simplistic. Muscle strength might be more relevant in older adults and dynamic conditions (Paillard, 2017; see also Point 2), and tendon properties may also play a role (Onambele et al., 2006). Moreover, I would expect the peripheral nervous system, which tunes information from the central nervous system, to be also likely involved in this process. Anthropometry (e.g. changes in body weight induced by bed rest) (Chiari et al., 2002; Hue et al., 2007) may also influence the biomechanics of postural balance control. I suggest that the authors either briefly discuss these aspects in dedicated paragraphs or more clearly acknowledge and justify their specific focus on neural and sensory factors, adjusting the title and text accordingly.

Reply: *We thank the reviewer for the kind words and raising this important point. We agree that the term "neuromuscular" comprises several further aspects which are not discussed in the present manuscript. The intention of the present topical review is to have a clear and specific focus on neural factors. Therefore, we adjusted the title to make the scope of our manuscript more. Further, all occurrences of "neuromuscular" were changed to "neuromotor".*

2) It is now well-established in the literature that static and dynamic postural balance control are not interchangeable and represent well-distinct paradigms (Ringhof & Stein, 2018; Rizzato et al., 2021). For instance, previous studies suggested employing dynamic rather than static tests to detect postural balance control impairments resulting from previous injury (Ross & Guskiewicz, 2004) or to study the impact of acute physical exercise on postural balance control (Marcolin et al., 2019). Moreover, the determinants of static and dynamic postural balance control potentially differ (Paillard, 2017). As the Authors mentioned both static and dynamic balance were assessed in previous bed rest studies (L113), this distinction should be explored better throughout the review.

Reply: *The reviewer raises a valuable point. We agree and added more information were necessary to highlight which studies assessed dynamic and static postural balance control to be more transparent for the readers. Additionally, we added a few sentences to paragraph 6 to highlight the relevance of both static and dynamic assessments.*

3) In some paragraphs (e.g., 3.1.3; 3.2), the authors should not only describe the specific physiological alterations occurring with bed rest but also explain how these alterations could affect postural balance control in this context. See also my comment on sleep behaviour in the 'Specific Comments' section.

Reply: Thank you for your valuable feedback. The entire section has been revised to ensure a clearer connection between the physiological alterations induced by bed rest and their potential impact on postural balance control. Additional explanations have been incorporated into sections 3.1.3 to explicitly discuss how changes in electrocortical activity, sensorimotor input, and cerebrospinal fluid dynamics may influence postural stability. Additionally, the paragraph on EEG and sleep has been removed as it was not a direct fit for the discussion. We appreciate your insightful suggestions, which have helped to improve the clarity and relevance of our manuscript.

4) Based on the literature, is it possible to propose a potential 'time course' for these alterations during bed rest? Among the different proposed mechanisms, which occur earlier and which develop at a later stage?

Reply: Thank you for your thoughtful comment. While we provide a summary of results in Table 1, the studies included in the table assess different parameters of balance control (e.g., COP displacement, equilibrium scores, functional mobility tests) and various postural modalities (e.g., eyes open/closed, dynamic head tilts, different support surfaces). Due to this variability, it is not possible to establish a straightforward and uniform time course of balance changes/recovery. As indicated in Tables 1-3, measurement conditions, parameters, variables, paradigms and metrics are diverse among the study portfolio, that there is no credible and serious approach to summarizing the results properly and presenting them in a metric.

Initially, we considered presenting a time course similar to what we previously reported for knee strength and muscle mass recovery in Marusic et al. (2021, *Journal of Applied Physiology*). However, the heterogeneity in balance assessment methods and the multifactorial nature of postural control make such a structured timeline more challenging to define. Instead, we have focused on identifying general trends in balance impairment and recovery while acknowledging inter-study differences.

Marusic, U., Narici, M., Simunic, B., Pisot, R., & Ritzmann, R. (2021). Nonuniform loss of muscle strength and atrophy during bed rest: a systematic review. *Journal of Applied Physiology*, 131(1), 194-206.

5) Adding a table summarising the designs and findings of the studies described in Section 2 would improve accessibility for readers by providing a detailed overview of the investigations on postural balance control alterations during bed rest. The table should include elements such as study design, sample size, methods employed, and the type of balance assessed, offering a clearer snapshot of the relevant research, as this is not described in detail in the text.

Reply: We thank the reviewer for this comment. We agree that tables summarizing the designs and findings of the included studies in Section 2 is necessary and important to gain detailed overview of the current state of research. As suggested, we have prepared 3 tables including study design, sample size, measured variables and study results for static postural conditions (Table 1), dynamic postural conditions (Table 2) and anticipatory reactive responses (Table3). We have decided to present the effects of bed rest on biomechanical parameters of balance control in a narrative and not in a numeric (i.e. mean and std dev)

manner. The reason is that the presentation of numeric parameters is as manifold as the parameters are and include boxplots, CI, means, medians, modelling approaches, SE, SE etc. The narrative presentation of the adaptation allows the readership of the JP to capture the key results of the numerous studies executed on this topic easily.

6) The Authors used interchangeably the terms "balance control", "postural equilibrium", "postural control", and "balance". I recommend the Authors to keep the terminology consistent throughout the manuscript. To the best of my knowledge, the term "postural balance control" is considered the most complete and is generally employed by experts in the field (E.g. Marcolin et al., 2022).

Reply: *Thank you for this important comment. We agree that there should be a more consistent use of terminology. As suggested, we used "postural balance control" throughout the entire manuscript now.*

Specific Comments:

Title page: Author order and corresponding author in the manuscript do not coincide with the submission information on the website. Please clarify this aspect.

Reply: *Thank you. The order of the authors and corresponding author is correct as stated in the manuscript. We will adjust this accordingly in the submission system.*

L118: Is there any study exploring the effects of dual-task on postural balance control following bed rest? If yes, this should be highlighted in this paragraph.

Reply: *We agree that dual-task conditions would be highly interesting in those populations. However, to the best of our knowledge, no studies are available which describe the effects of dual task on postural balance control in populations undergoing bed rest.*

L143: Specify that the higher prevalence of falls was tested experimentally and not observed in a "real world" scenario.

Reply: *Thank you for mentioning this point. We added this as suggested.*

L167-182: Observations from this paragraph are exclusively derived from very long-term bed rest studies. The Authors are invited to acknowledge that results may differ with short-term bed rest.

Reply: *Thank you. We added this as suggested.*

L240-249: Are the changes in EEG patterns during sleep relevant to postural balance control?

If so, the authors should explain the connection. If not, I suggest removing this paragraph to maintain focus.

Reply: *Thank you for your comment. The paragraph discussing EEG patterns during sleep has been removed, as it was not directly relevant to postural balance control.*

L276-277: To the best of my knowledge, the opposite trend was often observed in the disuse literature. A previous bed rest case study found an increased Hmax/Mmax ratio (Duchateau, 1995). Similar findings were also reported in other disuse models in humans (Lundbye-Jensen & Nielsen, 2008; Seynnes et al., 2010), and in related animal studies (De-Doncker et al., 2006; Islamov et al., 2011). I understand this review is not the place for a methodological discussion on the topic, but I suggest the Authors consider reporting these findings are not univocal.

Reply: *Thank you for this comment. We acknowledge that the findings are not univocal among different disuse models. Therefore, we have included additional information regarding the univocal results found in humans.*

L301: The Authors may consider also adding to this paragraph this recent manuscript on changes in retinal venular vessel diameters: (Saloň et al., 2023).

Reply: *Thank you for your suggestion. We have incorporated the findings from Saloň et al. (2023) into Section 3.3.2 to highlight the reduction in retinal venular diameter during bed rest and its potential implications for ocular perfusion and visual function.*

L365: Have the Authors considered expanding this paragraph with a comparison of bed rest with other disuse models, such as unilateral lower limb suspension or immobilisation? While these unilateral models can also disrupt postural balance control (Elam et al., 2022), the mechanisms may differ due to the lack of fluid shift and systemic effects, as well as the lower likelihood of impacting vision and causing structural brain changes. While I do not consider this a requirement, as the authors have motivated their focus on bed rest, it could add an interesting dimension to the manuscript.

Reply: *We appreciate the reviewer's comment. Our team discussed this issue extensively and even drafted the paragraph below. However, after careful consideration, we found that incorporating it into the review article disrupts somehow the overall focus. In case we want to keep it, the subtitle should be changed to "Comparing disuse models" or similar. If the reviewer thinks that this should be incorporated, we can do it in the R2? The paragraph is provided below and would fit from the row 471 onwards:*

..., as compared to more than 50 articles related to HDTBR. While bed rest is a widely used model for studying systemic effects of physical inactivity, other disuse paradigms such as unilateral lower limb suspension and immobilization provide insight into more localized neuromuscular adaptations. Studies on short-term unilateral immobilization, such as Elam et al. (2022), have demonstrated that even two weeks of lower limb immobilization can significantly impair postural control, particularly in older individuals,

due to reductions in muscle power and neuromuscular coordination. However, unlike bed rest, these models lack the fluid shifts, vestibular alterations, and widespread neural deconditioning associated with prolonged recumbency. The absence of these systemic effects in unilateral models suggests that bed rest-induced postural instability arises not only from musculoskeletal deconditioning but also from impairments in sensory integration and vestibular processing. This distinction highlights why HDTBR, which simulates the cephalad fluid shifts of microgravity, may result in more profound disruptions in balance control compared to unilateral immobilization models. In contrast, 4° to 8° HDTBR introduces additional physiological stress...

L450: The author contribution section seems more referred to an original research article rather than a narrative review. Please amend it accordingly.

Reply: Changes as suggested, thank you.

L712 (Figure 2): In order to enhance the Figure's readability and make it more schematic, I recommend the Authors change, where possible, verbs with symbols. E.g. "↓ directional cognition" instead of "decreased directional cognition".

Reply: Thank you for this suggested. We adjusted the Figure 2:

L712 (Figure 2): Are there some parameters changing with long but not short-term bed rest? If yes, the Authors are invited to specify this in the figure or figure caption.

Reply: We thank the reviewer for pointing this out. In connection to our response (4), we acknowledge that there is currently no definitive time course that can be extracted from the available studies. In the tables, we have summarized the parameters that were assessed during different durations of bed rest, but we cannot conclusively determine which parameters change exclusively with long-term versus short-term bed rest. Rather, the studies report findings based on specific time points without a consistent trend across durations. To address this, we have clarified in the figure caption that the parameters were assessed at different bed rest durations, without implying a strict temporal progression. We appreciate the reviewer's suggestion and have revised the caption accordingly.

References:

Chiari L, Rocchi L & Cappello A (2002). Stabilometric parameters are affected by anthropometry and foot placement. *Clin Biomech* 17, 666-677.

De-Doncker L, Kasri M & Falempin M (2006). Soleus motoneuron excitability after rat hindlimb unloading using histology and a new electrophysiological approach to record a neurographic analogue of the H-reflex. *Exp Neurol* 201, 368-374.

Duchateau J (1995). Bed rest induces neural and contractile adaptations in triceps surae. *Med Sci Sports Exerc* 27, 1581-1589.

Elam C, Hvid LG, Christensen U, Kjær M, Magnusson SP, Aagaard P, Kall LB & Suetta C (2022). Effects Of Aging On Changes In Muscle Power And Postural Control After Immobilization And Re-training. *J Musculoskelet Neuronal Interact* 22, 486-197.

Hue O, Simoneau M, Marcotte J, Berrigan F, Doré J, Marceau P, Marceau S, Tremblay A & Teasdale N (2007). Body weight is a strong predictor of postural stability. *Gait Posture* 26, 32-38.

Islamov RR, Mishagina EA, Tyapkina O V., Shajmardanova GF, Eremeev AA, Kozlovskaya IB, Nikolskij EE & Grigorjev AI (2011). Mechanisms of spinal motoneurons survival in rats under simulated hypogravity on earth. *Acta Astronaut* 68, 1469-1477.

Lundbye-Jensen J & Nielsen JB (2008). Immobilization induces changes in presynaptic control of group Ia afferents in healthy humans. *J Physiol* 586, 4121-4135.

Marcolin G, Panizzolo FA, Biancato E, Cognolato M, Petrone N & Paoli A (2019). Moderate treadmill run worsened static but not dynamic postural stability of healthy individuals. *Eur J Appl Physiol* 119, 841-846.

Marcolin G, Supej M & Paillard T (2022). Editorial : Postural Balance Control in Sport and Exercise. *Front Physiology* 13, 12-13.

Onambele GL, Narici M V. & Maganaris CN (2006). Calf muscle-tendon properties and postural balance in old age. *J Appl Physiol* 100, 2048-2056.

Paillard T (2017). Relationship between Muscle Function, Muscle Typology and Postural Performance According to Different Postural Conditions in Young and Older Adults. *Front Physiol* 8, 1-6.

Ringhof S & Stein T (2018). Biomechanical assessment of dynamic balance: Specificity of different balance tests. *Hum Mov Sci* 58, 140-147.

Rizzato A, Paoli A, Andretta M, Vidorin F & Marcolin G (2021). Are Static and Dynamic Postural Balance Assessments Two Sides of the Same Coin? A Cross-Sectional Study in the Older Adults. *Front Physiol* 12, 1-8.

Ross SE & Guskiewicz KM (2004). Examination of static and dynamic postural stability in individuals with functionally stable and unstable ankles. *Clin J Sport Med* 14, 332-338.

Saloň A, Çiftci GM, Zubac D, Šimunič B, Pišot R, Narici M, Fredriksen PM, Ngwenchi B, Chungag N, Sourij H, Šerý O, Zalaudek KS & Steuber B (2023). Retinal venular vessel diameters are smaller during ten days of bed rest. *Sci Rep* 1-7.

Seynnes OR, Maffiuletti NA, Horstman AM & Narici M V. (2010). Increased H-reflex excitability is not accompanied by changes in neural drive following 24 days of unilateral lower limb suspension. *Muscle Nerve* 42, 749-755.

REQUIRED ITEMS

- Please include an Abstract Figure file, as well as the Figure Legend text within the main article file. The Abstract Figure is a piece of artwork designed to give readers an immediate understanding of the Review Article and should summarise the main conclusions. If possible, the image should be easily 'readable' from left to right or top to bottom. It should show the physiological relevance of the Review so readers can assess the importance and content of the article. Abstract Figures should not merely recapitulate other figures in the Review. Please try to keep the diagram as simple as possible and without superfluous information that may distract from the main conclusion of the Review. Abstract Figures must be provided by authors no later than the revised manuscript stage and should be uploaded as a separate file during online submission labelled as File Type 'Abstract Figure'. Please ensure that you include the figure legend in the main article file. All Abstract Figures will be sent to a

professional illustrator for redrawing and you may be asked to approve the redrawn figure before your paper is accepted.

Reply: We appreciate the editorial guidance on including an Abstract Figure. As requested, we have prepared a simplified Abstract Figure that visually summarizes the main conclusions of our review, ensuring it is easily readable and highlights the physiological relevance of the topic. The Abstract Figure file has been uploaded separately under 'File Type: Abstract Figure,' and the corresponding figure legend has been added to the main article file.

The Journal of
Physiology

- Please upload separate high quality figure files via the submission form.

Reply: Thank you, we uploaded them in high quality.

- Author profile(s) must be uploaded via the submission form. Authors should submit a short biography (no more than 100 words for one author or 150 words in total for two authors) and a portrait photograph of the two leading authors on the paper. These should be uploaded and clearly labelled together in a Word document with the revised version of the manuscript. Any standard image format for the photograph is acceptable, but the resolution should be at least 300 DPI and preferably more. A group photograph of all authors is also acceptable, providing the biography for the whole group does not exceed 150 words.

Reply: we provide our CVs below:

Ramona Ritzmann, PhD, specializes in neurophysiological stimulation techniques to study motor control and neuroplasticity. She has worked with the European Space Agency on gravity and deconditioning research and led biomechanics and performance diagnostics at a Swiss sports medicine clinic. Since 2021, she has been with the AO Foundation, becoming Head of Clinical Operation in 2022, overseeing clinical research and trials.

Uros Marusic holds a B.S. in Biomedical Engineering and a Ph.D. in Kinesiology. He is the Head of the Laboratory for Kinesiological Research and leads the Slovenian Mobile Brain/Body Imaging Lab (SloMoBIL) at ZRS Koper. His research focuses on muscle-brain crosstalk, particularly in aging, neurodegeneration (e.g., Parkinson's disease), and space physiology.

Dear Professor Ritzmann,

Re: JP-TR-2025-285668R1 "**Neuromotor changes in postural control following bed rest**" by Uros Marusic, Christoph Centner, Luke Hughes, Janice Waldvogel, and Ramona Ritzmann

We are pleased to tell you that your paper has been accepted for publication in The Journal of Physiology.

Authors should note that it is too late at this point to offer corrections prior to proofing. Major corrections at proof stage, such as changes to figures, will be referred to the Editors for approval before they can be incorporated. Only minor changes, such as to style and consistency, should be made at proof stage. Changes that need to be made after proof stage will usually require a formal correction notice.

Yours sincerely,

Richard Carson
Senior Editor
The Journal of Physiology

P.S. - You can help your research get the attention it deserves! Check out Wiley's free Promotion Guide for best-practice recommendations for promoting your work at www.wileyauthors.com/eeo/guide. You can learn more about Wiley Editing Services which offers professional video, design, and writing services to create shareable video abstracts, infographics, conference posters, lay summaries, and research news stories for your research at www.wileyauthors.com/eeo/promotion.

IMPORTANT NOTICE ABOUT OPEN ACCESS: To assist authors whose funding agencies mandate public access to published research findings sooner than 12 months after publication, The Journal of Physiology allows authors to pay an Open Access (OA) fee to have their papers made freely available immediately on publication.

You can check if your funder or institution has a Wiley Open Access Account here: <https://authorservices.wiley.com/author-resources/Journal-Authors/licensing-and-open-access/open-access/author-compliance-tool.html>.

EDITOR COMMENTS

Reviewing Editor:

We thank the author for their response to the prior suggestions. The work is far more relevant to the physiological underpinnings of ageing & frailty than previously.

The figures in particular are easily digestible.

The conclusion has a good narrative clarity, and the writing seems accessible to the non-niche reader.

REFeree COMMENTS

Referee #1:

The Authors have done an excellent job addressing my concerns. Key amendments include the addition of useful tables, a sharper focus on neuromotor changes in relation to postural balance control, and a clearer distinction between dynamic and static postural balance control.

Regarding the suggestion to expand the discussion to other disuse models (e.g., unilateral limb immobilisation), if the Authors feel this would disrupt the focus of the paper, its inclusion is not necessary.

I suggest that, at the proof stage, the Authors specify in the main text that the tables are reported in the Supplementary Information.

Congratulations on a nice and relevant review.